# The Frontal Aslant Tract and Supplementary Motor Area Syndrome: Moving towards a Connectomic Initiation Axis

**DOI:** 10.3390/cancers13051116

**Published:** 2021-03-05

**Authors:** Robert G. Briggs, Parker G. Allan, Anujan Poologaindran, Nicholas B. Dadario, Isabella M. Young, Syed A. Ahsan, Charles Teo, Michael E. Sughrue

**Affiliations:** 1Department of Neurosurgery, University of Oklahoma Health Sciences Center, Oklahoma City, OK 73104, USA; gbriggs023@gmail.com (R.G.B.); Parker-allan@ouhsc.edu (P.G.A.); 2Brain Mapping Unit, Department of Psychiatry, University of Cambridge, Cambridge CB2 1TN, UK; ap2057@cam.ac.uk; 3Doctoral Program, The Alan Turing Institute, British Library, London NW1 2DB, UK; 4Department of Neurosurgery, Prince of Wales Private Hospital, Sydney 2031, Australia; nbd37@rwjms.rutgers.edu (N.B.D.); isabella@cingulumhealth.com (I.M.Y.); syed.a.ahsan@student.unsw.edu.au (S.A.A.); charlie@neuroendoscopy.info (C.T.); 5Rutgers Robert Wood Johnson Medical School, New Brunswick, NJ 08901, USA

**Keywords:** tractography, SMA syndrome, glioma surgery, frontal aslant tract, connectomics, parcellation, neurosurgery, neuro-oncology

## Abstract

**Simple Summary:**

Connectomics enables us to map whole brain networks that can be applied to operative neurosurgery to improve neuro-oncological outcomes. Damage to the superior frontal gyrus during frontal lobe surgery is thought to induce supplementary motor area (SMA) syndrome in patients. However, network-based modeling may provide a more accurate cortical model of SMA syndrome, including the Frontal Aslant Tract (FAT). The aim of our study was to retrospectively assess if surgical tractography with diffusion tensor imaging (DTI) decreases the likelihood of SMA syndrome. Compared to patients who underwent surgery preserving the SFG (*n* = 23), patients who had their FAT and SMA networks mapped through DTI and subsequently preserved were less likely to experience transient SMA syndrome. Preserving the FAT and SMA improves functional outcomes in patients following medial frontal glioma surgery and demonstrates how network-based approaches can improve surgical outcomes.

**Abstract:**

Connectomics is the use of big data to map the brain’s neural infrastructure; employing such technology to improve surgical planning may improve neuro-oncological outcomes. Supplementary motor area (SMA) syndrome is a well-known complication of medial frontal lobe surgery. The ‘localizationist’ view posits that damage to the posteromedial bank of the superior frontal gyrus (SFG) is the basis of SMA syndrome. However, surgical experience within the frontal lobe suggests that this is not entirely true. In a study on *n* = 45 patients undergoing frontal lobe glioma surgery, we sought to determine if a ‘connectomic’ or network-based approach can decrease the likelihood of SMA syndrome. The control group (*n* = 23) underwent surgery avoiding the posterior bank of the SFG while the treatment group (*n* = 22) underwent mapping of the SMA network and Frontal Aslant Tract (FAT) using network analysis and DTI tractography. Patient outcomes were assessed post operatively and in subsequent follow-ups. Fewer patients (8.3%) in the treatment group experienced transient SMA syndrome compared to the control group (47%) (*p* = 0.003). There was no statistically significant difference found between the occurrence of permanent SMA syndrome between control and treatment groups. We demonstrate how utilizing tractography and a network-based approach decreases the likelihood of transient SMA syndrome during medial frontal glioma surgery. We found that not transecting the FAT and the SMA system improved outcomes which may be important for functional outcomes and patient quality of life.

## 1. Introduction

Frontal lobe surgery has long been performed with respecting local neuroanatomy and avoiding eloquent brain areas. Traditionally, intra-axial cortical neurosurgery has primarily focused on avoiding motor deficits [1]. However, there are arguably more adverse outcomes after inadvertent frontal lobe transgressions such as leaving the patient unresponsive or unable to engage with the world around them. The challenge for neurosurgeons when operating on the frontal lobe is that the functional anatomy is poorly understood [2], often resulting in unpredictable post-operative functional outcomes [3]. Thus, there is a dire clinical need to surgically approach the frontal lobes with a neuroscientific framework and consider spatially distributed functional brain networks governing frontal lobe function. With rapid advances in imaging and high-throughput technologies [4], neurosurgery is entering a new era of connectome-based surgical targeting [5] and whole-brain operative planning [6], enabling us to better map the brain and begin to potentially improve neuro-oncological outcomes. 

Supplementary motor area (SMA) syndrome is a uniquely neurosurgical problem characterized by hemiparesis and mutism [3]. The current dogma argues that SMA syndrome is induced after surgery in or near the posteromedial bank of the superior frontal gyrus. While most patients improve over time, some patients do not and recovery trajectories are non-uniform and unpredictable [7]. Moreover, in our surgical experience with frontal lobe tumours, we have noted that patients experience varying degrees of mutism and hemiplegia from operations that do not even involve the canonical SMA [8]. This suggests that our localised cortical model of SMA syndrome is incomplete and insufficient to avoid this problem. 

Recently, we described the prefrontal cognitive initiation ‘axis’ [6]. In brief, the Default Mode Network connected via the cingulum, the Salience Network connected via the frontal aslant tract (FAT), and the SMA form a structural chain in the medial frontal lobe. The integrity of this initiation ‘axis’ is crucial for the brain to internally model goal initiation, appropriately dialogue with the Multiple Demand (MD) system [9], and ultimately execute goal-directed behaviour. In this study, we aimed to determine if surgical tractography of the frontal aslant tract (FAT) in ‘initiation axis’ decreases incidence of SMA syndrome. 

## 2. Materials and Methods

### 2.1. Patient Population and Data Collection

Patients who underwent surgery at the University of Oklahoma for an infiltrating Grade II, III, or IV tumour were analysed from the senior author’s database. Only tumours located within the posteromedial frontal lobe, with a of majority tumour tissue within or deep to the posterior SFG or medial portion of the middle frontal gyrus (MFG), were included in the analyses. Clinical records, hospital charts, and imaging studies were reviewed through the last available follow-up. Patients who were not seen at least three months after surgery were noted as lost to follow-up.

### 2.2. Defining SMA Syndrome

SMA syndrome in patients was defined as the presence of a contralateral hemiparesis (arm weakness roughly equal to leg weakness) with mutism occurring immediately after surgery that could not be explained by ischemic injury, injury to the precentral gyrus, or transgression of the corticospinal fibres seen on diffusion tensor imaging (DTI) when comparing pre-operative and post-operative images. Resolution of SMA syndrome was defined as functional improvement in both speech and motor deficits at any time point following surgery. Any degree of elicited difficulty initiating speech or moving the contralateral arm and leg was considered non-resolution of SMA syndrome. We sub-classified all medial frontal lesions based on whether they primarily threatened the SMA in its traditional position along the medial bank of the posterior SFG, or whether they primarily threatened the FAT along its medial portions (i.e., the superolateral parts of the SFG and medial MFG). 

### 2.3. Operative Technique

The first fundamental comparison being made in this study is between the traditional operative technique in the medial frontal lobe versus a modified technique to account for the FAT and the initiation axis. 

#### 2.3.1. Traditional Surgical Approach

During the initial cases in this series, the primary goal was to avoid injuring the primary motor cortex, the adjacent descending corticospinal fibres, the anatomic region of the SMA, as well as its corticospinal connections. We attempted to avoid resecting the traditional cortical location of the SMA in the posterior medial SFG whenever possible. We performed all cases with the patient awake and undergoing subcortical motor mapping using a combined naming and motor task [9]. We performed DTI fibre tractography in all cases, with the goal of preserving the anatomic SMA and its connections to the primary motor cortex posteriorly and the deep brain structures inferiorly. Lateral to this basic plane, we focused on making a coronal cut in the deep white matter as proximal to the precentral gyrus as motor mapping would allow. If the patient developed signs of SMA syndrome during this part of the procedure, we immediately stopped the dissection and only resected tumour anterior to this cut. We attempted to preserve the superior longitudinal fasciculus in all cases using DTI to guide the lateral extent of the dissection.

#### 2.3.2. Modified Approach to Avoid the FAT

After studying the anatomy of the FAT in 2015, the senior author (M.E.S.) began to perform all posterior frontal glioma resections with intra-operative neuronavigation, now including with the FAT as highlighted in the Medtronic Stealthviz Software (Medtronic, Dublin, Ireland). The identification of the FAT is demonstrated in Figure 1. Tumour location varied in relation to the FAT, but was often observed in front of the FAT (Figure 2). Regardless, we planned to resect these tumours by making coronal cuts parallel to the FAT so as to avoid resecting this tract.

### 2.4. Outcome Assessment

Patients underwent a full neurological examination by the senior neurosurgeon immediately after surgery and within three months of follow-up in the neurosurgical clinic. Last known follow-up time was defined as the day of the operation to last known follow-up in clinic. Tumour volumes were calculated using pre-operative, contrast-enhanced T1-weighted magnetic resonance imaging scans when possible or T2-weighted MRI scans if lesions were non-enhancing [10]. Post-operative MRIs were completed within 48 h of surgery. Extent of resection was calculated by subtracting post-operative tumour volume from pre-operative tumour volume divided by pre-operative tumour volume. Final pathological diagnosis was confirmed by the in-house neuropathologist. 

### 2.5. Statistical Analysis of Clinical Outcomes

All between-group comparisons for continuous variables were performed using the independent samples t-test. All between-group comparisons for categorical variables were performed using the Chi-Squared Test. Results were considered statistically significant if the *p*-value was less than or equal to 0.05. 

### 2.6. Construction of Network Maps and Anatomy of the “Command and Control” Axis

#### 2.6.1. Literature Search Strategy

Literature searches for all relevant coordinate-based fMRI studies related to attention, language, auditory, and motor processing were completed using BrainMap Sleuth 2.4 (http://www.brainmap.org/sleuth/, accessed on 10 May 2020) as well as PubMed and Google Scholar if no fMRI studies could be identified in the BrainMap fMRI database [11]. Studies were included in our analysis if they met the following criteria: (1) peer-reviewed publication, (2) task-based fMRI study related to the default mode network, salience network, executive control network, or supplementary motor area, (3) based on whole-brain, voxel-wise imaging, (4) including standardized coordinate-based results in the Talairach or Montreal Neuroimaging Institute (MNI) coordinate space, and (5) including coordinates for at least one healthy human control cohort. Only coordinates from healthy subjects were utilized in our analysis.

#### 2.6.2. ALE Generation and Identification of Relevant Cortical Regions

We used BrainMap Ginger ALE 2.3.6 to extract the relevant fMRI data to create an anatomic likelihood estimation (ALE) based on the coordinate data for each network [12,13,14]. All coordinates were exported to GingerALE in the MNI coordinate space. We subsequently performed a single study analysis using cluster-level interference (cluster level of 0.05, threshold permutations of 1000, uncorrected *p*-value of 0.001). The ALE coordinate data were displayed on an MNI-normalized template brain using the Multi-Image Analysis GUI (Mango) 4.0.1 (http://ric.uthscsa.edu/mango/, accessed on 10 May 2020). Using ROIs constructed using the Connectome Workbench command line interface from our previous work, we assessed parcellations for inclusion in each network if the parcellation and ALE overlapped [15,16,17].

#### 2.6.3. Tractography 

After identifying the relevant cortical parcellations of each network, fibre tractography between parcellations was performed in DSI Studio (http://dsi-studio.labsolver.org, accessed on 10 May 2020) using publicly available brain imaging from the Human Connectome Project (http://humanconnectome.org, accessed on 10 May 2020, release Q3). Tractography was performed individually with ten randomly chosen adult subjects. A multi-shell diffusion scheme was used, with *b*-values of 990, 1985, and 2980 s/mm^2^. Each *b*-value was sampled in 90 directions. The in-plane resolution was 1.25 mm. The slice thickness was 1.25 mm. The diffusion data were reconstructed using generalized q-sampling imaging [18]. The diffusion sampling length ratio was 1.25. 

All reconstructions were performed in MNI space using a region of interest (ROI) approach to initiate fibre tracking from a seeded region. Voxels within each ROI were automatically traced with a maximum angular threshold of 45°. When a voxel was approached with no tract direction or a direction greater than 45°, the tract was halted. Tracks with length shorter than 30 mm or longer than 800 mm were discarded. In some instances, exclusion ROIs were placed to exclude spurious tracts or tracts inconsistently represented across individuals. Tracts were considered real between parcellations if they could be identified consistently in five or more subjects.

## 3. Results

### 3.1. Patient Population

In total, there were 45 patients included in this study. Of these, 23/45 patients (51%) were part of the control group who underwent resection using the traditional approach without preserving the FAT. The remaining 22/45 patients (49%) were part of the treatment group and underwent tumour resection with the addition of FAT mapping with DTI tractography. 

The demographics of these patients are summarized in Table 1. The two groups did not differ by age, gender, extent of resection, tumour laterality, tumour size, or tumour grade (*p*-value > 0.05). In both groups, approximately two-thirds of patients (67%) had tumours directly invading the region of the SMA proper, while approximately one-third of patients (33%) had tumours located in the postero-medial SFG, involving the FAT. Note that in patients undergoing surgery prior to our attempts to preserve the FAT, the relationship between the tumour and the FAT was determined retrospectively by post-processing the pre-operative DTIs used during the initial surgery.

### 3.2. Surgical Outcomes

When planning our surgeries to prevent cutting or disrupting the fibres of the FAT, we found that the rate of temporary post-operative SMA syndrome was significantly reduced compared to operating without preserving the FAT. In the FAT preservation group, 2/22 (8.3%) patients developed temporary SMA syndrome compared to 11/23 (47%) patients whose FAT was not preserved (*p* = 0.003). In contrast, no statistically significant difference was detected between these groups with respect to the development of permanent SMA syndrome after surgery. In the FAT preservation group, 0/22 (0%) patients developed permanent SMA syndrome compared to 3/23 (13%) patients whose FAT was not preserved (*p* = 0.232). While this result was not statistically significant, no patients in the FAT preservation cohort developed permanent SMA syndrome. These data are summarized in Table 2.

When assessing the occurrence of SMA syndrome relative to tumour location, we found no statistically significant differences (*p* > 0.05) between groups (Table 3). In patients presenting with tumours occupying the SMA proper, 8/29 (28%) patients developed temporary SMA syndrome compared to 5/16 (31%) patients who presented with tumours involving only the FAT (*p* = 1.000). Similarly, 1/29 (3%) patients presenting with tumours occupying the SMA developed permanent SMA syndrome compared to 2/16 (13%) patients presenting with tumours involving only the FAT (*p* = 0.285). When considering the effect of FAT preservation in these two groups, we found that maintaining the connections of the FAT was associated with a significantly reduced incidence of temporary SMA syndrome following surgery in patients who present with lesions involving the FAT alone (*p* = 0.03). No significant differences in the rates of temporary or permanent SMA syndrome were detected between patients presenting with tumours within the SMA proper when comparing between groups for which the FAT was preserved versus cut (*p* > 0.05). Similarly, no significant difference in the rate of permanent SMA syndrome was detected between patients presenting with tumours involving the FAT alone when comparing between groups for which the FAT was preserved versus cut. The results of this sub-group analysis are summarized in Table 4.

The Appendix A contains a detailed illustration of our prior work on the connectomic anatomy of the medial frontal lobe. 

## 4. Discussion

In this report, we present evidence that SMA syndrome may be an avoidable surgical complication if a conscious effort is made to avoid transgressing the FAT. Importantly, this can be achieved without limiting the extent of tumour resection. Given the inability of patients with SMA syndrome to initiate motor or speech activity, it is likely that it results from inadvertently damaging medial frontal lobe brain networks required to convert internal goals and signals into executable action plans.

Our work with network modelling suggests that the FAT may be the principle pathway linking the SMA to premotor areas and area 44 (the canonical Broca’s area) [19,20]. As demonstrated in this study, the FAT also links the principle hubs of the salience network, a cingulo-insular-opercular network that has been shown to mediate the transition between internal and external mental states [21]. Given the inability of patients with SMA syndrome to initiate motor or speech activity, it seems reasonable to hypothesize that SMA syndrome results from disruption to the interconnections of the networks used to convert internal goals and signals into executable action plans.

### 4.1. Challenges with Certainty

As with other methods to avoid neurological decline following surgery, there is inherent uncertainty in the claim that preserving the FAT helps avoid the development of SMA syndrome. Tractography has some uncertainty, especially in the setting of peritumoral edema [22,23]. This uncertainty increases when one claims they preserved a tract during surgery, as post-operative edema makes tractography near the margins of the tumour almost impossible to perform [23], especially considering the clinical DTI paradigms we use and the neuroanatomic distortion that occurs in the immediate post-operative period. Studying this endpoint using delayed diffusion MRI after the edema has resolved is one possible solution. However, this approach can also be problematic given the potential for loss to follow-up in the post-operative period, adjuvant radiotherapy-related changes within the white matter, and the development of tumour recurrence. Thus, we cannot know with complete certainty that the FAT was preserved or that this is the definitive reason why SMA syndrome was observed less frequently in these cases. However, given that the only substantial change made between the cohorts included in this analysis involved mapping and displaying the FAT during surgery in order to preserve the tract, preservation of the FAT leading to a reduction in SMA syndrome represents the best present hypothesis to explain the results.

Still, alternate explanations need to be considered. For example, by avoiding the FAT, we were in effect avoiding the canonical SMA. This represents a confounding operative factor for which we cannot correct. However, SMA syndrome was observed in patients in this series who did not have tumours occupying the SMA proper suggesting that SMA syndrome and similar motor initiation problems are not limited to operating in the traditional anatomic SMA. Another possibility is that by aiming to respect the SMA, we were effectively being less aggressive during surgical resection, limiting injury to the SMA and other motor cortical areas. However, while we observed a decrease in the extent of resection when purposefully attempting to preserve the FAT, this reduction was relatively minor and not statistically significant.

### 4.2. Onco-Functional Balance for Surgically Induced SMA Syndrome in Glioma Patients 

Transient SMA syndrome can occur in more than half of patients following SMA resection [24,25], a finding consistent with the data presented here. However, the rapid recovery and limited permanent neurologic deficit in these patients is often considered acceptable for a successful surgery [7], such as to eliminate severe epilepsy [26].

It may be unwise, though, to apply such reasoning to glioma surgery. Glioma surgery involves subtle nuances that need to be considered to maximize the onco-functional balance. For instance, resection of the medial frontal lobe may be necessary to eliminate medically intractable seizures, but glioma surgery involved gradations in tumour resection and the extent of SMA resection correlates with the amount and severity of post-operative neurologic deficits [24,25]. Furthermore, the severity of these deficits following additional resection in tumour patients can further disrupt the start of adjuvant therapy, worsening the underlying syndrome [27]. Thus, preserving the FAT to promote a faster recovery despite a trade-off in extent of resection may represent the optimal form of treatment for patients with tumours occupying the posterior medial frontal lobe.

It is important to also note that while others have argued that the post-operative deficits are transient in nature, we found that not preserving the FAT resulted in permanent deficits in 13% of patients, which is far from trivial. Elsewhere, reports of surgically induced SMA syndrome are permanent in 7% - 20% of patients, suggesting that our findings are within line of previous studies [24,28]. In tumour patients with already decreased median overall survival, the added complication of immobility can further deteriorate a patients quality of life with increased risk of death [29,30].

### 4.3. Moving towards the Idea of Preserving the “Initiation” Axis

Despite the fact that transgressing through some parts of the frontal lobe is generally well-tolerated, disruption of frontal lobe function can lead to life-altering neurologic problems. Akinetic mutism, abulia, and other medial frontal lobe syndromes can be thought of as difficulties with the initiation of spontaneous, internally motivated actions, primarily manifested by a lack of self-initiated activity.

Previously, we reported our results with a technique for removing anterior butterfly gliomas which reduced the rate of akinetic problems by preserving the anterior cingulate gyrus [31]. Additional study of the connectomic anatomy of this area has expanded our understanding of the probable role of the anterior cingulate cortex and its neighbouring cortices in promoting action initiation leading to an anatomic concept of an initiation system in the neocortex [6] (see Appendix A). This axis is not as simple an anatomic concept as the “motor cortex”, as the deficits caused by damaging parts of the axis are less rigidly related. For example, some patients can tolerate a unilateral cingulate resection, while others cannot [31]. Instead, this model serves to provide a framework for understanding the relevant structural and functional properties of the brain when neurosurgeons must operate within the medial frontal lobe.

### 4.4. Techniques for Operating around the Initiation Axis

After demonstrating the utility of mapping the FAT and initiation axis, we now discuss important technical nuances for operating in this corridor. 

Even if one aims to stay within the tumour during glioma surgery, aggressive resection of these tumours often involves creating divisions within the cerebrum. At a minimum, this means cutting around the border of the tumour into healthy brain. Thus, it is helpful to think of the resection as a set of subcortical disconnections, with the goal being to minimize unnecessary destruction of salvageable networks. Intra-operative use of fibre tractography is extremely helpful in this regard as it can be used for planning cuts around critical white matter tracts and corresponding brain networks [32,33,34]. Without intraoperative fibre tractography, there would be no landmarks to structure the surgical manoeuvres necessary to resect the tumour in the subcortical white matter.

It is important to note that, unlike cutting the corticospinal tract, unilateral injury to the initiation axis does not uniformly result in abulia or akinesis. Our previous experience suggests that some fraction of patients can tolerate unilateral resection of some parts of this axis [31]. However, given that the incidence of abulia or akinesis may be as high as 30–40% [31], we have adopted the view that one should not ignore the anatomy of the initiation axis, unless doing so is absolutely critical for tumour control. Otherwise, the results may be unsatisfactory for patients. In our experience, it is rare that preserving the axis and obtaining adequate tumour removal are mutually exclusive with proper technique.

#### 4.4.1. Coronal Cuts within the Motor System and along the Frontal Aslant Tract

When operating near eloquent aspects of the motor system, cuts must be made so as to prevent causing permanent motor deficits. In general, the surgeon can protect this system by dividing the subcortex parallel to the direction of the motor fibres and making coronal cuts parallel to the precentral or postcentral gyri.

Adopting the same surgical principle as outlined for divisions within the motor cortex, our team has found that a coronal cut that begins at the level of the frontal operculum and ends in the frontal horn can preserve the FAT. 

#### 4.4.2. Sagittal Cuts within the Cingulum

The callosal fibres that arise along the length of the medial hemispheres make the same characteristic bend around the cingulum bundle as they project to analogous parts of the parasagittal brain, i.e., SFG to SFG, paracentral lobule to paracentral lobule [35]. This important fact suggests that it is possible to preserve the cingulate system while removing the corpus callosum. This should not be surprising, given that we have been safely cutting through the corpus callosum for years, and have been entering the frontal horn transcortically for years through the forceps minor [36].

It is important to think of anterior callosal tumours as frontal tumours which involve the corpus callosum. Ideally, these tumours should be resected through the middle frontal gyrus [37], which allows the neurosurgeon to avoid the DMN and SN entirely during surgery. To achieve this, the neurosurgeon needs to identify the cingulate cortex from the interior medial surface of the lobe. In addition to using the cingulate sulcus as a key landmark, intra-operative mapping should be used to reveal when the surgeon should deviate laterally to avoid cutting through cingulum bundle on the way to the ventricle.

#### 4.4.3. Tumours Which Split the FAT and Primary Motor Cortex

While the majority of the posterior frontal tumours that we have since encountered push the FAT backwards towards the primary motor cortex, we have encountered several cases where the tumour splits the FAT and the primary motor cortex (Figure 2a). In Figure 2b, the tractography demonstrates that the FAT is located anterior to the tumour, and that the tumour is splitting the FAT and primary motor cortex. This tumour was successfully resected using awake motor mapping and planned surgical cuts to preserve the FAT. The patient did not experience any muscle weakness, abulia, or other initiation problems after surgery, despite a relatively aggressive resection near the anatomic SMA extending into the postcentral gyrus.

Reorganization of this low grade oligodendroglioma can be posited as part of the reason this was possible in this and several other similar patients we have treated. However, it seems that in patients like this, it is not necessary to maintain the majority of connections between the SMA and most of the primary motor cortex as most of those possible pathways should have been cut with this surgery, and yet there was no obvious consequence in this patient. Thus, it seems that the best hypotheses are that either (a) the FAT tractography was accurate and that preserving it is important, (b) the FAT tractography is inaccurate, the tract has been long inactivated, and the patient has adapted, and/or (c) the SMA talks to the motor cortex principally through its basal ganglia circuit. We do not believe that U-fibre connections can explain the lack of SMA syndrome in this patient as most the remaining U-fibre connections are linking the residual medial posterior Superior frontal gyrus to the leg portion of the paracental lobule, which does not strike us as sufficient to initiate internally driven motor plans. It seems likely to us that both hypotheses (a) and (c) are true to some extent; however, given the goal of most surgeons is to avoid cutting the corticospinal fibres, which was always our goal before and after the addition of FAT preservation, and merely avoiding these fibres did not eliminate the SMA syndrome, while avoiding the FAT did. Thus, we hypothesize that the FAT is the key reason this patient did not develop motor problems from our surgery.

### 4.5. Future Work in Cognition

The addition of clinical neuropsychology has benefited the neurooncological practices in terms of improved patient quality of life through monitoring patient functions such as cognition [38]. However, the majority of focus on SMA syndrome has leaned towards preventing patient deficits of motor and speech function rather than higher-order executive functions, such as in working memory [39]. Given that the SMA and FAT is part of the prefrontal cognitive initiation ‘axis’ and the drastic functional improvements patients demonstrated in this study by preserving the FAT and SMA tracts, it is plausible that regressions of these tracts may lead to multi-network disturbances causing global executive dysfunction. Damage to the FAT has been correlated with attentional difficulties and working memory performance [40,41]. Damage to the executive component of working memory is needed for complex functions of reasoning and learning, which is demonstrated in SMA lesion studies that produce cognitive deficits other than language related skills, such as in attention processing and arithmetic processes [39]. Preoperative neuropsychological testing could provide additional information on the effect of preserving the FAT and SMA on neurocognitive functions following medial frontal lobe glioma surgery. Furthermore, in particular, it will be important to examine connectivity with the anterior cingulate, which has been associated with cognitive control [42].

## 5. Conclusions

In this study, we present data suggesting that mapping the FAT with tractography can mitigate the likelihood of post-operative SMA syndrome. By consciously avoiding the transgression of the FAT during medial frontal lobe surgery, we are able to preserve the communication between multiple large-scale brain networks critical for goal-directed behaviour. This work provides a growing foundation in favour of ‘connectomic’ surgery in ultimately improving neuro-oncological outcomes. With emerging surgically relevant connectomic platforms on the rise [43], this work adds support towards ‘connectomic surgery’ in ultimately improving neuro-oncological outcomes.

## Figures and Tables

**Figure 1 cancers-13-01116-f001:**
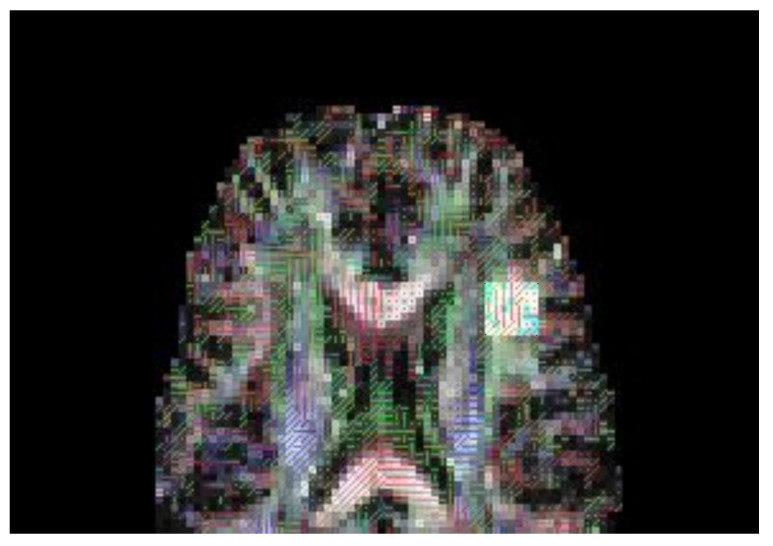
The FAT was identified by placing a ROI seed, seen as a white square, near the cranio-caudal fibres which terminate anterior to the arcuate fasciculus frontal projections at the level of the inferior frontal gyrus (IFG).

**Figure 2 cancers-13-01116-f002:**
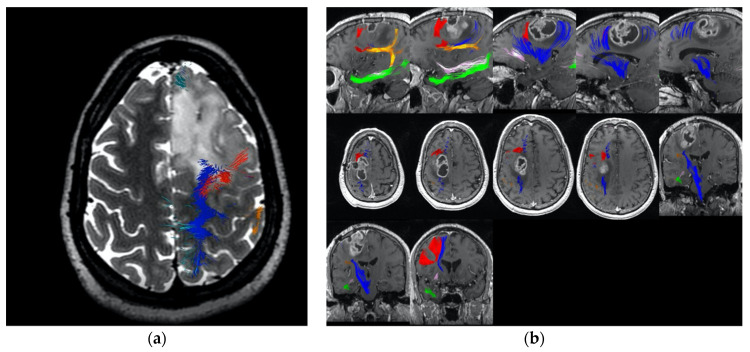
These figures demonstrate the relative position of the major white matter tracts relevant to SMA. The corticospinal tract is in blue, the FAT in red, the arcuate fasciculus is in orange and the cingulum fibres are in mauve. (**a**) Demonstrates the tumour pushing the FAT fibres behind towards the primary motor cortex; (**b**) Demonstrates the tumour between the FAT and the primary cortex, pushing the FAT anteriorly.

**Table 1 cancers-13-01116-t001:** Patient demographics.

	No FAT Preservation	FAT Preservation	Chi-Squared Test
Number of Patients (N)	23	22	
Gender (M/F)	13/10	11/11	*p* = 0.76
Age	44 ± 0.2	49 ± 2.6	*p* = 0.06
Tumour Laterality (L/R)	13/10	16/6	*p* = 0.35
Tumour Size (cc)	48 ± 1.4	37 ± 8.6	*p* = 0.22
Extent of Resection			*p* = 0.23
<80%	2/23 (9%)	3/22 (14%)	
80–90%	1/23 (4%)	5/22 (23%)	
91–99%	5/23 (22%)	5/22 (23%)	
100%	15/23 (65%)	9/22 (40%)	
Tumour Grade			*p* = 0.29
Grade 2	9/23 (39%)	4/22 (18%)	
Grade 3	4/23 (17%)	6/22 (27%)	
Grade 4	10/23 (43%)	12/22 (55%)	
Location			*p* = 1.000
SMA Proper	15/23 (65%)	14/22 (64%)	
Near the FAT	8/23 (35%)	8/22 (36%)	

**Table 2 cancers-13-01116-t002:** Occurrence of supplementary motor area syndrome based on preservation of the FAT in brain tumour patients.

	No FAT Preservation	FAT Preservation	Chi-Squared Test
Number of Patients (N)	23	22	
Temporary SMA Syndrome	11/23 (47%)	2/22 (8.3%)	*p* = 0.003
Permanent SMA Syndrome	3/23 (13%)	0/22 (0%)	*p* = 0.232

**Table 3 cancers-13-01116-t003:** Occurrence of supplementary motor area syndrome based on the anatomic location of tumours in brain tumour patients.

	In SMA Proper	FAT Involved Only	Chi-Squared Test
Number of Patients (N)	29	16	
Temporary SMA Syndrome	8/29 (28%)	5/16 (31%)	*p* = 1.000
Permanent SMA Syndrome	1/29 (3%)	2/16 (13%)	*p* = 0.285

**Table 4 cancers-13-01116-t004:** Sub-Group Analysis of the Occurrence of SMA Syndrome Based on Preservation of the FAT in Tumours Occupying the Supplementary Motor Area versus the FAT.

	In SMA Proper			FAT Involved Only		
	No FAT Preservation	FAT Preservation	Chi-Squared Test	No FAT Preservation	FAT Preservation	Chi-Squared Test
Number of Patients (N)	14	15		8	8	
Temporary SMA Syndrome	6/14 (43%)	2/15 (13%)	*p* = 0.11	5/8 (63%)	0/8 (0%)	*p* = 0.03
Permanent SMA Syndrome	1/14 (7%)	0/15 (0%)	*p* = 0.48	2/8 (13%)	0/8 (0%)	*p* = 0.49

## Data Availability

The data presented in this study is available on request from the corresponding author. The data is not publicly available due to patient confidentiality.

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
