# Peer review of "The Frontal Aslant Tract and Supplementary Motor Area Syndrome: Moving towards a Connectomic Initiation Axis"

_cancers, 2021, doi:10.3390/cancers13051116_

Round 1

Reviewer 1 Report

The article depicts an important issue in brain oncological surgery. The SMA syndrome is an important matter in pre-motor cortex tumors, as so far, there is no validated method to avoid it. The intraoperative recognition of the FAT as demonstrated by the authors is indeed timely and relevant to reduce the postoperative risk of SMA syndrome. The authors should be commended for presenting evidence that SMA syndrome may be an avoidable surgical complication if a conscious effort is made to avoid transgressing the FAT without limiting the extent of tumor resection. The paper shades light to the paramount ‘shift’ towards brain connectomics in neuroncology.

In particular the FAT preservation had an impact in both temporary and permanent SMA syndromes. My only concern deals with the relatively elevated permanent SMA syndromes  (up to 13%) in cases without FAT preservation. I would like authors to comment on this aspect.

The article is well written, the topic is relevant and I recommend publication.

Author Response

Please see attachment below for a point-by-point response to the reviewer's comments. Thank you.

Reviewer 2 Report

This paper is very interesting. I strongly recommend this should be accepted. This paper deserves neurosurgeons, neuroradiologists and neurologists to have a deep read and think over it.

The writing is excellent, and the introduction is very good, very attracting.

Since the content is like a combination of “retrograde analysis”, and “review”, in the introduction the authors introduced the Default Mode Network, the Salience Network, and the Multiple Demand (MD) system, however, the authors did not go further to analysis their patients’ psychological status, such as any elicited disturbance in DMN, the salience network or the MD in their patients, whether with or without FAT preservation.

Table 4 is a weak point in this paper. Most readers will notice that in the permanent SMA syndrome. This is no significant differences between groups with or without FAT preservation no matter in the in SMA proper or FAT involved only. I may suggest the authors to do another analysis to explore any group with SMA with or without (any) neurological or psychological impairments, and you may find something interesting and statistically significant. 

Author Response

(The authors gave the same response as above.)
